# A Novel Multi-Step Global Mechanism Scheme for n-Decane Combustion

**DOI:** 10.3390/e25101389

**Published:** 2023-09-28

**Authors:** Shaozhuan Xiong, Yantian Bi

**Affiliations:** 1Research Center of Combustion Aerodynamics, Southwest University of Science and Technology, Mianyang 621000, China; 2Institute of Mechanics and Aerospace, Southwest University of Science and Technology, Mianyang 621000, China

**Keywords:** n-decane, mechanism reduction, chemical kinetic simulations, sensitivity analysis

## Abstract

Based on the directed relation graph with error propagation (DRGEP) reduction method, a detailed mechanism consisting of 119 species and 527 reactions for n-decane was simplified. As a result, a skeletal mechanism comprising 32 species and 73 reactions was derived. Subsequently, the quasi-steady state approximation (QSSA) reduction method was employed to further simplify the skeletal mechanism, resulting in a reduced mechanism with 18 species and 14 global reactions. A comparison between the reduced mechanism, skeletal mechanism, and detailed mechanism revealed that the reduced and skeletal mechanisms successfully replicated the combustion characteristics of the detailed mechanism under a range of initial conditions. These models can be credibly incorporated into large-scale combustion simulation, serving as a solid foundation for enhancing computational efficiency.

## 1. Introduction

In recent years, the numerical simulation of aviation fuel has gained significant importance in optimizing engine designs for performance, efficiency. Aviation fuel is composed of hundreds to thousands of hydrocarbon compounds, including paraffins, cycloparaffins, aromatics, and alkenes, with a wide range of molecular weights. Unfortunately, detailed chemical kinetic mechanisms that accurately describe the combustion of many components in aviation fuels are currently unavailable. Therefore, to simulate the combustion of aviation fuels, simplified model fuels (surrogates) consisting of a few representative hydrocarbons must be utilized. Detailed kinetic mechanisms offer comprehensive and accurate representations, however they tend to be large and time-consuming when employed in practical three-dimensional computational fluid dynamics (CFD) simulations. As a result, mechanism reduction becomes essential to efficiently apply such extensive mechanisms in CFD simulations of complex flow fields.

Mechanism reduction has been extensively studied, and numerous methodologies have been developed over the past few decades [1,2,3,4,5,6,7,8,9,10,11,12,13,14]. These methods can be broadly categorized into two major groups. The first category is skeletal reduction, which involves eliminating unimportant species and reactions from the detailed mechanism. Several methods fall into this category, including sensitivity analysis [1], level of importance (LOI) [2], directed relation graph (DRG) [3], and other DRG-based techniques such as DRG with error propagation (DRGEP) [4], path flux analysis (PFA) [5], revised DRG [6], DRG-aided sensitivity analysis (DRGASA) [7], DRGEP with sensitivity analysis (DRGEPASA) [8], as well as flux path tree (FPT) [9]. The second category is based on time scale analysis, which approximates fast processes with algebraic equations. The main global reduction methods in this category currently include quasi-steady state approximation (QSSA) [10], the computational singular perturbation (CSP) method [11,12], and intrinsic low-dimensional manifold (ILDM) [13,14].

China RP-3 kerosene is a crucial hydrocarbon fuel widely used in aeroengine because its high energy density and stable thermodynamic properties. Moreover, given that the physical and chemical properties of n-decane closely resemble those of RP-3 kerosene, n-decane (C_10_H_22_) has been selected as a single-component surrogate fuel to develop a detailed chemical kinetics mechanism for RP-3 aviation kerosene. Over the past few decades, numerous studies have focused on modeling the combustion properties of n-decane. The n-decane mechanisms presented in previous studies are summarized in Table 1.

Dagaut et al. [15] initially used n-decane as a surrogate to describe their experimental results of TR0 kerosene oxidation in a jet-stirred reactor. Westbrook and co-workers [16] presented a comprehensive kinetic reaction mechanism for C8-C14 n-alkanes, the mechanism for n-decane included 940 species and 3878 reactions, and demonstrated good agreement between simulation and experimental data. Bikas and Peters [17] developed a chemical kinetic mechanism for n-decane, comprising 67 species and 600 elementary reactions. The mechanism was validated against many types of experimental results from shock tubes, jet-stirred reactors, and premixed laminar flames, spanning low to high temperatures, yielded satisfactory results.

In an effort to better represent the ignition delay at low temperatures in shock tubes, Honnet et al. [18] extended n-decane mechanism to a full temperature region based on the mechanism of Bikas and Peters [17]. This new mechanism was consisted of 119 species and 527 reversible reactions, and its prediction shows all-right agreement with the experimental results at a series of conditions. Zeng et al. [19] developed a detailed kinetic model for the pyrolysis and oxidation of n-decane, incorporating 234 species and 1452 reactions, which was tested against previous experimental data within intermediate and high temperature regime. Liu et al. [20] generated a mechanism with 388 species and 2226 reactions specifically for n-decane. The ignition delay times predicted by this mechanism exhibited good agreement with experimental data under different pressures and temperatures. Other recent attempts to provide detailed mechanisms for alkanes, including n-decane, have utilized kinetic model generators such as EXGAS [21] and MOLEC [22]. Unfortunately, detailed chemical kinetic mechanisms involve hundreds to thousands of chemical species. Due to their large size, the computational cost required for running these detailed mechanisms becomes prohibitive when applied to practical three-dimensional simulations of engine combustion.

Therefore, significant efforts have been made to develop reduced and skeletal mechanism of n-decane. Sun et al. [5] developed a reduced n-decane mechanism derived from a high-temperature n-decane mechanism consisting of 121 species [23,24]. They employed the reduction methods of PFA and DRG to obtain reduced mechanisms with 54 and 55 species, respectively. Based on a decoupling methodology, Chang et al. [25] proposed a skeletal mechanism for n-decane with 40 species and 141 reactions. Wang et al. [26] developed an improved path flux analysis (IMPFA) with multi-generations method for mechanism reduction. They obtained a 63-species skeletal mechanism for n-decane using this approach. Zhong et al. [27] used the eigenvalue analysis reduction method to simplify a detailed n-decane mechanism of Honnet [18]. As a result, they derived a reduced mechanism with 38 species and 34 reaction steps, and the reduced mechanism have reproduced the combustion characteristics of n-decane. Yan et al. [28] proposed a simplified mechanism with 36 species and 62 elementary reaction steps. The validation of this mechanism against experimental data from the premixed flame of a Bunsen burner showed a good agreement. Xi et al. [29] employed an improved SA method to reduce the detailed mechanism of n-decane, and obtained a skeletal mechanism with 126 species. It predicted the variation trend of ignition delay time well, but overestimated the laminar flame speeds under some working conditions. However, incorporating reduced mechanisms into practical simulations of engine combustion remains challenging, especially for highly transient three-dimensional combustion processes.

**Table 1 entropy-25-01389-t001:** A list of previous studies on the n-decane mechanism.

References	Year	Species	Reactions
Dagaut [15]	1994	90	573
Westbrook [16]	2009	940	3878
Bikas [17]	2001	67	368
Honnet [18]	2009	119	527
Zeng [19]	2014	234	1452
Liu [20]	2012	388	2226
Sun [5]	2010	55/54	
Chang [25]	2013	40	141
Wang [26]	2016	63	
Zhong [27]	2014	38	34
Yan [28]	2016	36	62
Xi [29]	2020	126	

In order to meet the requirements of multi-dimensional numerical simulation for engine modeling and minimize the number of species, the present study has developed a comprehensive reduced reaction mechanism of n-decane. This reduced mechanism, consisting of 18 species and 14 global reactions, accurately predicts the combustion characteristics of n-decane across a wide range of operating conditions. The performance and accuracy of the reduced mechanism will be further evaluated in fundamental combustion simulations, along with the associated skeletal mechanism.

## 2. Theoretical Background

The process of mechanism reduction is illustrated in Figure 1. Initially, a large number of unimportant species and their corresponding reactions are eliminated from the detailed mechanism. First, DRG, DRGEP, PFA method is firstly employed to produce a set of different small-size mechanisms. This step is efficiently carried out using the DRGEP method for n-decane. Once the unimportant species are removed, further reduction is achieved by eliminating unimportant reactions based on the importance index derived from the CSP method. In this step, the number of species will not decrease but the complexity of the mechanism reduces.

Next, quasi steady-state (QSS) species are identified using CSP analysis.

An analytical solution is generated for the concentrations of the QSS species. Finally, the resulting reduced mechanism is rigorously validated against the detailed mechanism across a wide range of pressure, temperature, and equivalence ratios to ensure its accuracy.

### 2.1. Generalities about Chemical Mechanisms

In a combustion chemical mechanism, the thermodynamic information is mainly represented by the thermochemical quantities associated with each species, and the kinetic information is associated with the set of reactions that describe how these species react with each other. The thermodynamic parameters of a combustion reaction mechanism include the enthalpy of species, heat capacity, and entropy. The temperature dependent heat capacity, standard enthalpy and entropy are usually represented as two seven term form “NASA polynomials” [30]:(1)$$\frac{C_{P}^{0}}{R}=a_{1}+a_{2}T+a_{3}T^{2}+a_{4}T^{3}+a_{5}T^{4}$$
(2)$$\frac{H^{0}}{RT}=a_{1}+\frac{a_{2}}{2}T+\frac{a_{3}}{3}T^{2}+\frac{a_{4}}{4}a_{4}T^{3}+\frac{a_{5}}{5}a_{5}T^{4}+\frac{a_{6}}{T}$$
(3)$$\frac{S^{0}}{T}=a_{1}\ln T+a_{2}T+\frac{a_{3}}{2}T^{2}+\frac{a_{4}}{3}T^{3}+\frac{a_{5}}{4}a_{4}T^{4}+a_{7}$$

The rate coefficients of the combustion reaction mechanism are expressed using a modified three-parameter form of the Arrhenius equation:(4)$$k(T)=AT^{n}\exp(-E_{a}/RT)$$
where *A* is the pre-exponential factor, n is the temperature coefficient, *E_a_* is the activation energy, and *R* is the gas constant in J·K^−1^. It is important to note that each species has its own transport properties which are employed for the evaluation of gas-phase multicomponent viscosities, thermal conductivities, diffusion coefficients, and thermal diffusion coefficients.

### 2.2. DRGEP

The directed relation graph with error propagation (DRGEP) method, which has evolved from the original DRG method, was proposed by Pepiot-Desjardins and Pitsch [4]. The DRGEP method introduces a new interaction coefficient between species A and B which considers all possible direct and indirect couplings between the two species. The directed relation graph method for skeletal reduction was initially developed by Lu and Law [3]. This method utilizes a linear-time algorithm to identify species couplings and efficiently reduce large, detailed mechanisms.

However, the DRG method supposes the directed couplings of all species that are selected to be retained in the skeletal mechanism, which may not provide a comprehensive representation. Furthermore, contribution information about contribution strengths, as captured by the r_AB_ values, is lost by the binary truncation. To address these limitations, the DRGEP method is proposed to modify the DRG method by considering the propagation of error due to species removal down reaction pathways. In DRGEP, the interaction coefficient is computed as follows:(5)$$r_{AB}^{DRGEP}=\frac{\sum_{i=1,I}\left|v_{A,i}\omega_{i}\delta_{Bi}\right|}{\max\left(P_{A},C_{A}\right)}$$
with
(6)$$P_{A}=\sum_{i=1,I}\max\left(0,v_{A,i}\omega_{i}\right)$$
(7)$$C_{A}=\sum_{i=1,I}\max\left(0,-v_{A,i}\omega_{i}\right)$$
where *ν_A_*_,*i*_ represents the net stoichiometric coefficient of species A in the *i*th reaction and *ω_i_* denotes the net reaction rate for the *i*th reaction. *I* denote the total number of reactions in the detailed mechanism. If the *i*th elementary reaction involves species A, the value of *δ_Bi_* is 1; otherwise, it is 0. *P_A_* and *C_A_* represent the production and consumption rates of species A, respectively.

After calculating the interaction coefficients for all species pairs, a path-dependent interaction coefficient (PIC), *r_AB_*, *p*, represents the error propagation through a certain pathway and is defined as the product of intermediate interaction coefficients between target species A and species B through a certain path, *p*, in the directed graph. An overall interaction coefficient (OIC), *R_AB_*, is then defined as the maximum of all PICs between the target and each species of interest:(8)$$r_{AB,p}=\prod_{i=1}^{n-1}r_{S_{i}S_{i+1}}$$
(9)$$R_{AB}=\underset{\mathrm{all}\;\mathrm{paths}\;p}{\max}r_{AB,p}$$

Species B is considered to be unimportant and will be removed if *R_AB_* is below a user-specified threshold, εEP, and reactions consisting of unimportant components are also eliminated from the detailed mechanism. The optimal threshold is chosen by an iterative manner in this DRGEP implementation.

### 2.3. CSP to Remove the Unimportant Reactions

Lu and Law [31] have developed a method based on the CSP importance index to remove the unimportant reactions after eliminating the unimportant species. For this method, the importance index of a reaction is defined as:(10)$$I_{A,i}=\frac{\left|v_{A,i}\omega_{i}\right|}{\sum_{i=1,I}\left|v_{A,i}\omega_{i}\right|}$$
where *I_A_*_,*i*_ is the importance index of the *i*th reaction to the production rate of the *A*th species. If the importance index, *I_A_*_,*i*_, is below a user-defined threshold, ε, the *i*th reaction is considered to be unimportant for the reaction state.

### 2.4. Time-Scale Reduction with QSSA

In our study, the skeletal mechanism was next reduced using the QSSA method. However, two crucial challenges needed to be addressed when applying the QSSA method: the identification of QSS species and the efficient and reliable solution of the resulting algebraic equations [32]. QSS species are typically present in low concentrations and can be removed from the conservation equations without inducing significant errors among the remaining species. For the identification of QSS species, a criterion based on CSP has been proposed to provide a sufficient and necessary condition for a species to be QSS. It can be defined as follows:(11)$$J=\left(X_{fast},X_{slow}\right)\left(\begin{array}{cc} \Lambda_{fast} & 0\\ 0 & \Lambda_{slow} \end{array}\right){\left(Y_{fast}^{T},Y_{slow}^{T}\right)}^{T}$$
(12)$$X=\left(X_{fast},X_{slow}\right),Y={\left(Y_{fast}^{T},Y_{slow}^{T}\right)}^{T}$$The Jacobian matrix, denoted as ***J***, is defined as the derivative of the reaction rate vector, ***f***, with respect to the species concentration vector, ***c***. The chemical reaction kinetic model consists of K species and I elementary reactions. In this context, ***c*** represents the time rates of species concentration, while ***f*** represents the species concentrations themselves. The diagonalization matrix, **Λ**, is constructed by arranging the eigenvalues of the Jacobian matrix. Additionally, the matrices ***X*** and ***Y*** consist of the row and column eigenvectors of the Jacobian matrix, respectively. The criterion can be expressed as follows [33]:(13)$$D^{slow}=X_{slow}\cdot Y_{slow},\left|D_{i}^{slow}\right|<\varepsilon$$

***D*** represents the projection matrix onto the slow subspace and *ε* denotes a small threshold for the relative error tolerance. If the *i*th species satisfies Equation (13), it is considered to be a QSS species.

### 2.5. Brute Force Sensitivity Analysis

Brute force sensitivity analysis is performed through constant-volume adiabatic simulations. The percent sensitivity is defined according to the following equation [34]:(14)$$\%\mathrm{Sensitivity}=\frac{\tau \left(2k_{i}\right)-\tau \left(k_{i}\right)}{\tau \left(k_{i}\right)}\times 100\%$$
where *k_i_* represents the rate of reaction *i*, *τ*(2*k_i_*) denotes the ignition delay when the rate of reaction *i* is doubled, and *τ*(*k_i_*) represents the nominal value of the ignition delay. Therefore, a positive sensitivity value indicates that the ignition delay increases when the rate of reaction *i* is doubled.

## 3. Reduction Strategies

### 3.1. Skeletal Reduction

In our study, we selected Honnet’s n-decane mechanism [19] as original detailed mechanism, mainly due to its relatively small size and desirable predictive capability in combustion simulation. This mechanism, comprising 119 species and 527 reactions, is capable of accurately predicting various combustion characteristics over a wide range of conditions, including ignition delay, laminar flame speed, and extinction. Furthermore, it remains relatively small in size. Initially, we removed larger polycyclic aromatic hydrocarbon (PAH) species along with their associated reactions from the mechanism, except for benzene. This process resulted in an improved detailed mechanism consisting of 75 species and 373 reactions. The ignition delay and laminar flame speed calculations results using this mechanism and those of original detailed mechanism were compared to verify the accuracy of the improved detailed mechanism, and good agreement was found for the error limits considered (for further details on the validation results, please refer to the Appendix A). The improved detailed mechanism was then employed as the starting mechanism for the further mechanism reduction.

The newly obtained detailed mechanism, which originally consisted of 75 species and 373 reactions, underwent reduction to generate the skeletal mechanism using the DRGEP method. This was followed by further elimination of unimportant reactions. During the skeletal reduction process, the ignition delay time was chosen as the target parameter and validation simulations were performed over a wide range of initial conditions. These conditions included varying pressures from 1 to 10 atm, equivalence ratios ranging from 0.5 to 2.0, and initial temperatures spanning from 1000 K to 1600 K. The fuel, oxygen, and nitrogen were selected as the initial components for the mechanism reduction.

By adjusting the threshold value, ε, a series of skeletal mechanisms of different sizes were generated. Figure 2 illustrates changes in the relationship between the threshold values and the number of species in the resulting skeletal mechanisms, as well as the average error in predicting the ignition delay time. For very small threshold values (ε), the number of species in the skeletal mechanism closely resembled that of the detailed mechanism, and the average error in predicting the ignition delay was negligible. As the threshold value, ε, increased, the number of species in the skeletal mechanism decreased. Notably, Figure 2 depicted sudden changes or jumps in the average error of predicted auto-ignition delay around threshold values of ε = 0.15 and ε = 0.2.

After careful analysis, a threshold value of 0.20 was selected as the criterion, as it resulted in a reduced skeletal mechanism with 32 species. By eliminating unimportant elementary reactions from the aforesaid 32-species mechanism, a final skeletal mechanism consisting of 32 species and 73 reactions was obtained. The error in predicting the ignition delay remained within an acceptable range (30%) for this reduced skeletal mechanism.

Sensitivity analysis provides insights into the impact of reactions on the concentration of a specific species. To illustrate the key reactions in the skeletal mechanism influencing the ignition delay time during high-temperature combustion of n-decane, sensitivity analysis was performed for the ignition delay time in an n-decane/air mixture at different equivalence ratios, as shown in Figure 3. In the sensitivity analysis, a positive sensitivity value indicates that the corresponding reaction has an inhibiting effect on the ignition delay time, while a negative sensitivity value suggests a promoting effect. Figure 3 reveals that the most important reaction that promotes the ignition of n-decane across different equivalence ratios is consistently H + O_2_ = OH + O, which exhibits the largest inhibiting effect among all the reactions. Conversely, the reaction NXC_10_H_22_ + H = SXC_10_H_21_ + H exhibits the largest positive sensitivity value in the results derived from the skeletal mechanisms. Additionally, reactions involving HO_2_, H, OH, and CH_3_ radicals also play significant roles in the combustion properties of n-decane at high temperatures.

### 3.2. Reduced Reduction

The 32-species skeletal mechanism obtained earlier underwent further reduction using the QSSA method. The first step in QSSA-based reduction is to identify the QSS species. To achieve this, a criterion based on CSP was applied to the obtained reduced mechanism. This criterion was carried out at temperatures of 1000 K, 1200 K, 1400 K, and 1600 K, pressures of 1.0 atm and 10 atm, and equivalence ratios of 1.0 and 2.0. Based on this criterion, 14 species were identified as QSS species. These species are SXRO_2_, NXC_3_H_7_, PXCH_2_, PXC_7_H_15_, PXC_5_H_11_, SXC_10_H_21_, HCCO, HCO, TXCH_2_, O, C_2_H_3_, CH_3_O, H, and OH. Through the QSSA reduction, a final reduced mechanism with 18 species and 14 lumped global steps was obtained. When compared to the detailed mechanism, the maximum relative error in predicting ignition delay for the reduced mechanism is 18% and the average ignition error is 16.54%.

## 4. Reduction Validation

To further verify the performance of both the skeletal mechanism and the 18-species reduced mechanism, comprehensive comparisons were conducted between the simulation results obtained using the detailed mechanism and those obtained using the reduced mechanism. The direct comparisons encompassed various aspects and conditions, providing a thorough evaluation of the accuracy and predictive capability of the reduced mechanism in capturing the combustion behavior.

The ignition delay of a fuel/air mixture holds significant importance in *internal combustion* engines, as it heavily influences the combustion and emission characteristics. Therefore, in this section, the newly developed skeletal mechanism and reduced mechanism are validated against available experimental ignition delay data or corresponding simulation results obtained from the detailed mechanism. Modeling simulations were conducted with SENKIN using the CHEMKIN II program [35].

Figure 4 illustrates the calculated ignition delay times for n-decane/air mixtures using the skeletal mechanism, the reduced mechanism, and the detailed mechanism. The simulations cover a range of pressures from 1 atm to 50 atm and equivalence ratios from 0.5 to 2.0. It is evident that there is generally good agreement between the simulation results obtained with the skeletal mechanism, reduced mechanism, and detailed mechanism for most of the displayed conditions. However, slight over-prediction can be observed at low pressure and equivalence ratios ranging from 0.5 to 2.0.

Furthermore, as shown in Figure 4, we can conclude that the predicted results of the reduced mechanism closely align with those of the skeletal mechanism. This suggests that the reduced mechanism generated through the QSSA method can effectively replicate the combustion behavior obtained with the skeletal mechanism. Nevertheless, the discrepancy between the reduced and detailed mechanisms becomes more pronounced at lower initial temperatures, as shown in the three sets of figures. This discrepancy is attributed to the increased influence of temperature on the ignition delay time, particularly at high pressures or low initial temperatures. Consequently, the calculation error in the ignition delay time is relatively larger at lower temperatures. Upon further examination of the computational results, it is noted that the worst-case error remains below 30%, as stipulated by the restrictions imposed during the DRGEP and QSSA reductions.

To further illustrate the accuracy of species concentrations in auto-ignition, a comparison of the species profiles in auto-ignition simulations is presented in Figure 5. The simulations were conducted under the following conditions: an equivalence ratio (Φ) of 1.0, a pressure of 1.0 atm, and an initial temperature of 1000 K. From Figure 5, it is evident that the auto-ignition simulation results for the skeletal mechanism and the reduced mechanism can provide reliable predictions of the concentrations of key species compared to the detailed mechanism. Moreover, the results from the reduced mechanisms closely reproduce those of the skeletal mechanism, indicating that the reduced mechanisms accurately capture the behavior of the key species during auto-ignition.

In Figure 6, temperature profiles with different equivalence ratios calculated using the detailed mechanism, the skeletal mechanism, and the reduced mechanism are presented. The results of temperature profiles under various reaction conditions from both the detailed mechanism and skeletal mechanism exhibit similar agreement. Additionally, the temperature profiles calculated using the reduced mechanism also show good agreement with the results from the skeletal mechanism and the detailed mechanism. The validation results suggest that the discrepancies between the reduced mechanism and the detailed mechanism are primarily attributed to the errors inherent in the skeletal mechanism. Overall, the comparison of temperature variations is consistent across the entire range of equivalence ratios, with slightly larger errors observed for an equivalence ratio of 0.5. Sensitivity analysis conclusions in the Section 3.1 explain the impact of the equivalence ratios on the temperature profiles. When the equivalence ratio is small, near to 0.5, for example, the sensitivity value of the key reaction is large and the corresponding discrepancy between the reduced and detailed mechanisms becomes large.

Extinction is another important combustion property to validate chemical kinetic mechanisms and is also very important for kerosene combustion in jet engines. In Figure 7, temperature profiles as a function of residence time in a perfectly stirred reactor (PSR) under an equivalence ratio of 1 and pressure of 1 atm using the detailed mechanism, skeletal mechanism, and reduced mechanism are compared. It is evident that the skeletal mechanisms, reduced mechanism, and detailed mechanisms exhibit similar agreement when the residence time is greater than 0.01 s. However, a slight discrepancy in the computed temperature profiles is observed when the residence time is less than 0.01 s. In contrast, the 18-species reduced mechanism demonstrates good performance compared to the 32-species skeletal mechanism.

The present study developed a reduced n-decane mechanism consisting of 18 species and 14 lumped global reactions using the DRGER and QSSA methods, which strike a good balance between the size and accuracy of the mechanism. Then, the resulting reduced mechanism was rigorously validated against the detailed mechanism across a wide range of operation conditions to ensure its accuracy. These modeling simulation findings indicate that the reduced mechanisms, particularly the 18-species reduced mechanism, effectively capture the essential combustion behavior, such as the temperature profiles observed in the detailed mechanism. Aforementioned results show that as the size of the mechanism decreases, its performance in combustion simulations gradually becomes worse.

## Figures and Tables

**Figure 1 entropy-25-01389-f001:**
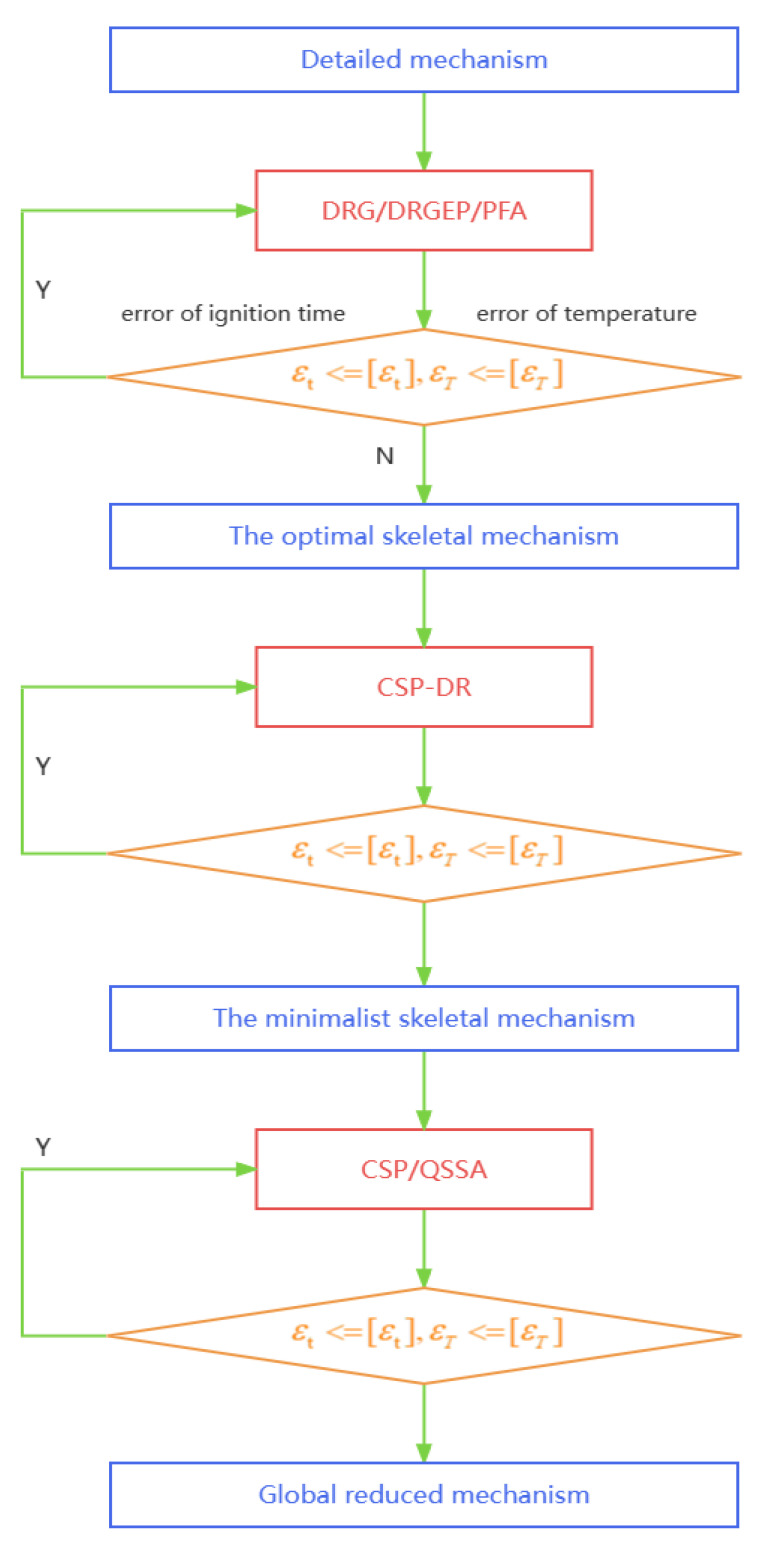
Schematic of the reduction procedures for the n-decane mechanism.

**Figure 2 entropy-25-01389-f002:**
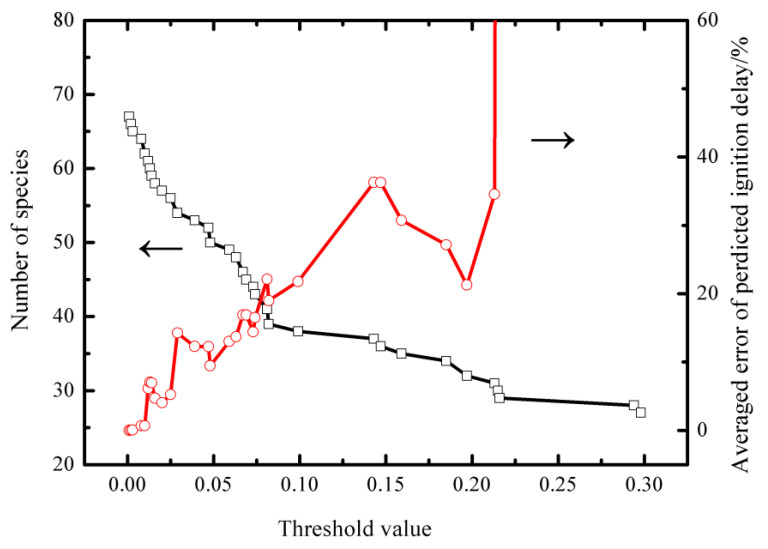
Number of species in the skeletal mechanism generated by DRGEP as a function of threshold values and the averaged error of the predicted auto-ignition delay of the skeletal mechanisms.

**Figure 3 entropy-25-01389-f003:**
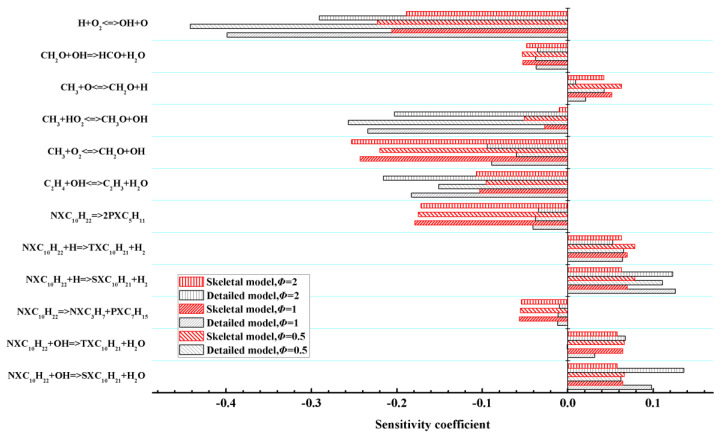
Sensitivity analysis for the ignition delay time in an n-decane/air mixture with constant pressure, a different equivalence ratio, and an initial temperature of 1200 K.

**Figure 4 entropy-25-01389-f004:**
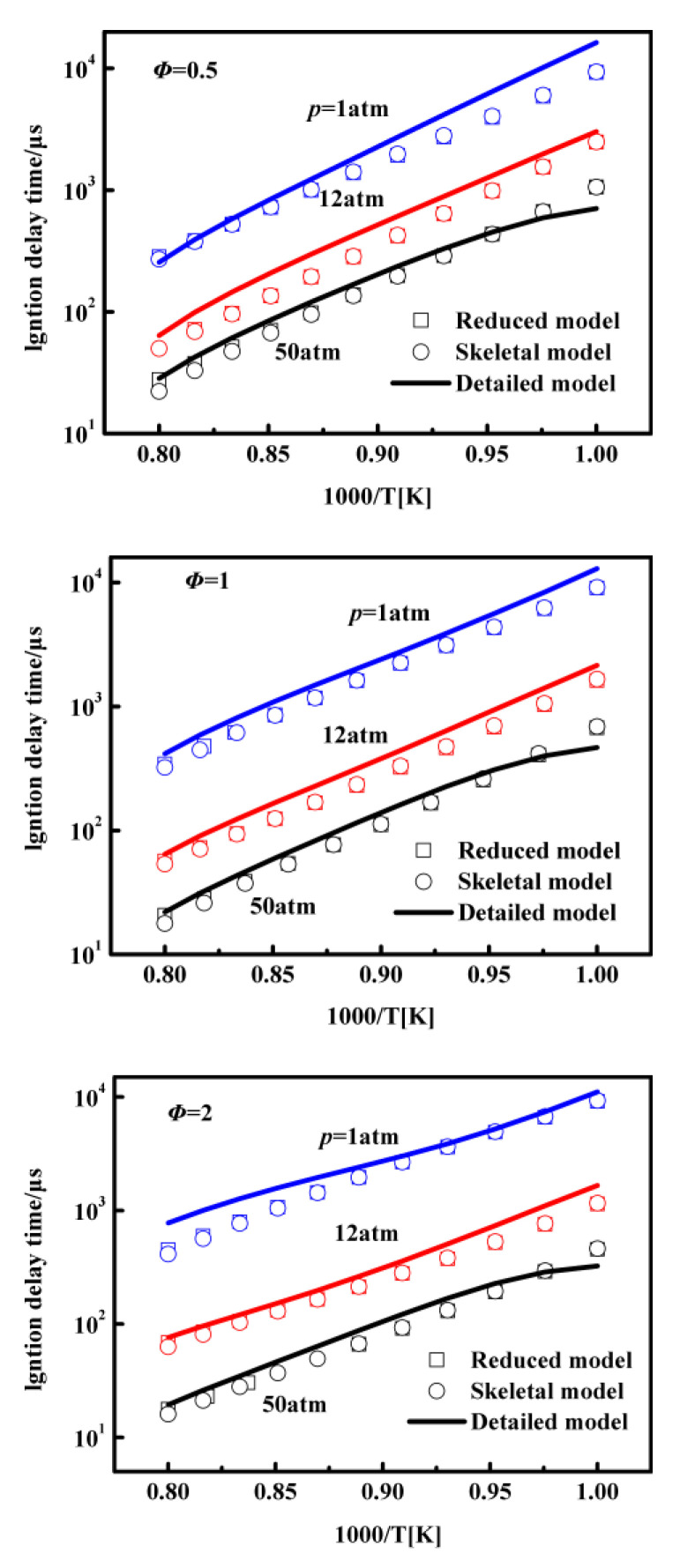
Ignition delay time of n-decane under equivalence ratios of 0.5–2.0 at 1–50 atm pressures in the detailed mechanism, the skeletal mechanism, and the reduced mechanism.

**Figure 5 entropy-25-01389-f005:**
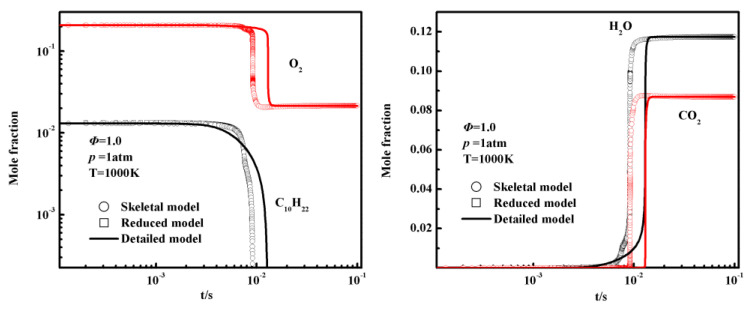
Species profiles in auto-ignition simulations of the detailed mechanism, the skeletal mechanism, and the 18-species reduced mechanism.

**Figure 6 entropy-25-01389-f006:**
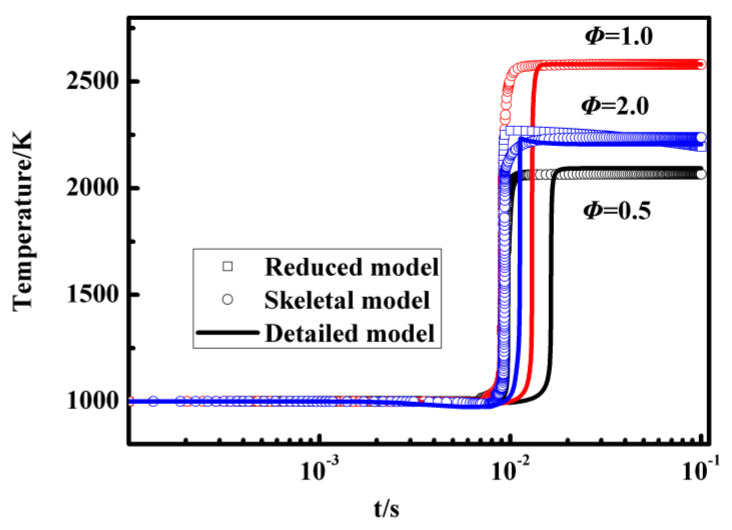
Temperature profiles in an auto-ignition simulation with constant pressure, different equivalence ratios, and an initial temperature of 1000 K calculated with the detailed mechanism, the skeletal mechanism, and the reduced mechanism.

**Figure 7 entropy-25-01389-f007:**
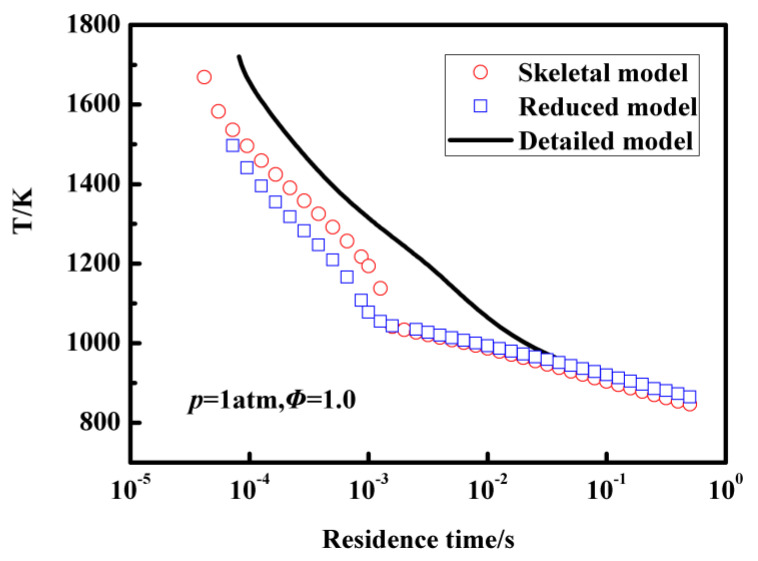
The temperature profiles of PSR at different residence times for the n-decane/air mixture, calculated with the detailed mechanism, 32-species skeletal mechanism, and 18-species reduced mechanism, respectively.

## Data Availability

No new data were created or analyzed in this study. Data sharing is not applicable to this article.

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
