# Peer review of "A Novel Multi-Step Global Mechanism Scheme for n-Decane Combustion"

_entropy, 2023, doi:10.3390/e25101389_

Round 1

Reviewer 1 Report

The manuscript presents a systematic reduction of detailed mechanism of n-decane, the methods are suitable, and the conclusions are suitably supported by results. I recommend publication in current manuscript.

Minor grammars should be checked.

Author Response

  Thank you very much for your approval of this manuscript, I have checked the manuscript and changed one grammatical error as you suggest.

Reviewer 2 Report

I am hesitant how to judge this paper. The title sounded interesting for me, but the article then disappointed me. I do not see much novelty in this paper. Maybe it is relevant for some one who faces the problem to simulate the combustion of this exact fuel. I acknowledge that the authors have given their reduced mechanism as supplemental material. However, in a more general perspective the reader does not learn much.

The theoretical background stays on the surface. The equations given look all familiar, but seem to be thrown in out of context. For instance, for the interpretation of the results it is not relevant how the thermodynamic properties are modeled. If it is meant to be a review of the methods, then there are too few equations, otherwise most of them could also be omitted.

In the results section, it seems to me that there is a systematic underestimation of the ignition delay time with the skeletal mechanism. I wonder why the methods do not produce a mechanism the equally deviates in both directions. Figure 6 also shows for me that the mechanism is not sensitive for the air/fuel ratio anymore. Maybe that is still good enough, but for me it is rather disappointing.

Unless other reviewers find the reduced mechanism useful for the community, I would suggest to reject this paper due to a lack of novelty.

Author Response

More details please see attachments.

Reviewer 3 Report

The manuscript aimed to create a new kinetic model of fuel combustion basing on the Directed relation graph with error propagation (DRGEP) reduction method. The proposed “reduced mechanism” had good consistence with the skeletal and detailed mechanisms. So, this manuscript can be accepted after some minor revisions.

1) In Fig.6 it was showed that the reduced model and skeletal model consisted with the detailed model only when the equivalence ratio is large, such as near to 2. How about the dependence of the accuracy of the created model on equivalence ratio.

      2) Where are the curves or symbols calculated from the reduced model in Fig. 5?

3) In Fig.7, it was shown that the accuracy of the skeletal model and the created reduced model only consisted with the detailed model at relative low temperature, especially in the temperature range of 1000-1200 K. So, it should be to reconsider the effect of temperature and to create a accurate model. 

English is well.

Author Response

More details please see attachments.

Round 2

Reviewer 2 Report

I am still not convinced about the novelty of this paper. However, since the other two reviewers find it interesting enough to be published, I will not object to its publication.

Reviewer 3 Report

the authours have sufficiently revised the manuscipt. It can be accepted for publication.